# CROSS-DOMAIN ATTENTION FOR TRANSFER LEARNING BETWEEN TABULAR DATA WITHOUT SHARED FEATURES

## ABSTRACT

Unlike images and text, transfer learning for tabular data is challenging due to heterogeneity in feature types, structures, and semantics across disparate application domains. Existing methods assume shared features across data tables to transfer knowledge, which may not hold across distinct application domains. To facilitate learning generalized context between domains without shared features, we propose a *data-agnostic* Cross-domain Attention Transfer Learning (CATTLE). CATTLE performs self-supervised pretraining of $key$, $value$, $query$ projection weights of a transformer-based model using source data. Pre-trained weights from selective attention layers are used in a separate transformer model to obtain generalized context for learning the target data via cross-attention. Our experiments on ten pairs of source-target data sets without shared features show that CATTLE is statistically and in terms of performance rank superior to nine state-of-the-art baselines, including traditional ML, deep tabular representation learning, and transfer learning methods proposed for tabular data. CATTLE demonstrates that the context for generalization can facilitate cross-domain transfer learning without shared features. A single tabular data set from an arbitrary domain is sufficient to learn a generalizable context for the target domain via cross-domain attention, addressing the challenge of pretraining models on many disparate and disjoint tabular data sets. CATTLE source code is available in [1].

## 1 INTRODUCTION

Beyond the capacity of traditional machine learning, transfer learning is a remarkable milestone in modern artificial intelligence. Transfer learning leverages knowledge from large, pre-trained models to adapt and generalize to domains with limited samples. The well-established notion of transfer learning in computer vision and language models stems from the properties of image and text data that are conducive to learning representations. Computer vision architectures, such as convolutional neural networks (CNNs) and vision transformers (ViTs), are successful in transfer learning because images provide generalizable, shared feature representations that effectively transfer knowledge across various domains Dosovitskiy et al. (2021). For example, large-vision models (e.g., ViT, VLM) trained with natural images can be finetuned using monochromatic medical images for disease classification tasks Lee et al. (2025).

In contrast, data structured in tables, also known as tabular data, contain a heterogeneous feature space Zhu et al. (2023), including numerical, categorical, ordinal, and textual data types. The transfer learning of data with a heterogeneous feature space remains challenging, where banking transaction tables do not share any plausible features with electronic health records data. Consequently, traditional feature engineering and empirical selection of classifier models are still in practice for tabular data. Therefore, no single learning method consistently stands out as the best model for a diverse set of tabular data Rabbani et al. (2024b). The diversity in tabular data sets introduces heterogeneity in the feature space, whereas image and text data exhibit a homogeneous distribution of pixels or words. Heterogeneity across disparate feature spaces is a significant barrier to learning generalizable knowledge necessary for transfer learning and building large foundation models.

---

[1]https://anonymous.4open.science/r/cattle-ED9F

To this end, state-of-the-art transfer learning methods rely on common features across tabular data sets, often within the same application domain, to ensure feature overlaps in features Wang & Sun (2024); Levin et al. (2023); Onishi et al. (2023); Nam et al. (2024). Transfer learning within the same domain assumes that the source and target data sets share similar feature distributions and contexts, enabling straightforward adaptation through model fine-tuning. In practice, tabular data sets across different domains often differ substantially in structure and feature distributions, with rarely any common features. This paper presents a novel transfer learning framework that effectively learns generalized context between tabular data sets without requiring a common feature space. The proposed method leverages cross-domain attention by selecting weights from the attention layer of a pretrained gated feature tokenizer transformer (gFTT) and embedding them into a target gFTT model for finetuning and downstream classification.

The paper is organized as follows. The remainder of this section provides motivation and contribution of this paper guided by the recent literature. Section 2 presents the preliminaries on attention-based computing and the proposed cross-attention framework for transfer learning. Section 3 discusses the data sets, the experiments, and the evaluation method. Section 4 compares the performance of our method with the state-of-the-art baselines and presents relevant ablation studies. Section 5 summarizes the main findings, provides insight into the results, and outlines limitations to suggest future work. The paper concludes in Section 6.

## 1.1 MOTIVATION

Tabular data are ubiquitous, thanks to the strong presence of relational databases in countless real-world applications, including banking, business, healthcare, and government enterprises. Despite the monopoly of deep learning, recent literature suggests that traditional machine learning (ML) still outperforms deep learning on tabular data Wang et al. (2021); Köhler et al. (2019); Smith et al. (2020); Shwartz-Ziv & Armon (2022); Borisov et al. (2022). However, traditional ML lacks some of the unique strengths of deep learning methods, including representation learning, incremental learning, and transfer learning. Therefore, complacency with ML performance limits our ability to build large data-driven models by adapting heterogeneous data sets from various domains. For example, many domains can afford to collect only limited data samples (e.g., rare diseases, expensive laboratory processes) for deep learning, which can benefit from transfer learning methods. However, transfer learning requires pre-trained models trained on large labeled data sets. While humans can recognize and annotate images and text needed to build pretrained models, it is neither trivial nor objective to manually assign a label to a feature vector in a tabular data set. Therefore, pretraining using a large volume of unlabeled tabular data for effective downstream finetuning and transfer learning is imperative, but challenging Levin et al. (2023).

## 1.2 LITERATURE REVIEW AND CONTRIBUTIONS

Decision tree-based learning, such as XGBoost Chen & Guestrin (2016), continues to dominate tabular data learning, despite the remarkable achievements of deep learning (DL) of image and text data Grinsztajn et al. (2022). Recent advances in deep tabular architectures, including TabNet Arik & Pfister (2021) and FT Transformer Gorishniy et al. (2021), outperform traditional machine learning models on some selective and large data sets with more than 10,000 samples. In practice, most tabular datasets have fewer than 1000 samples Rabbani et al. (2024a), which may not fully leverage deep representation learning. Furthermore, unlike for images and text, data augmentation and transfer learning solutions are not trivial for tabular data with limited samples Onishi et al. (2023). In the context of transfer learning, studies have shown that fine-tuning a pretrained model yields better performance on tabular data than using models without pretraining Cheng et al. (2024); Abrar & Samad (2022). However, similar pretraining and finetuning tasks are performed on subsets of the same data set. Similarly, TTNet Li et al. (2022) performs supervised pretraining followed by finetuning on another subset of the same data set. Similar pre-training solutions are not optimally designed for incrementally learning from disjoint tabular data sources to facilitate transfer learning.

Several recent studies have investigated transfer learning for tabular data using a pair of source and target data sets. The pretraining step in similar studies learns generalized representations on the source data. Pretrained models are then fine-tuned using target data with limited labeled samples. TransTab Wang & Sun (2024) performs transfer learning between two distinct source and target data

sets with several common features. An overlap in the feature space helps align the source and target feature embeddings, enabling effective knowledge transfer between tables. However, tabular data sets from disparate domains in the real world are unlikely to share common features. Levin et al. Levin et al. (2023) present a transfer learning method inspired by the need to predict rare diseases using limited patient samples. A transformer-based framework is pretrained on a large source data set with many patient samples for a multiclass disease classification task. The pre-trained model is fine-tuned using limited patient samples for a rare disease prediction task. The rare disease data set includes all features used in the upstream multiclass disease classification, as well as those unique to the rare disease. Feature similarities between source and target data within the same application domain may not reflect the broader context required for cross-domain knowledge transfer.

Recent studies present transfer learning between distinct domains and table schemas. The Cross-table Masked Modeling (CM2) framework Ye et al. (2024) is proposed to facilitate large-scale model pretraining using 2000 source data sets with varying schema, feature types, and distributions. However, the 16 target data sets used for downstream classification share features and domains with their source counterparts. XTab Zhu et al. (2023) is pre-trained by multiple source data sets using separate embedding layers for individual tables while simultaneously learning common features using a shared transformer. Despite multiple source data sets for pre-training, XTab struggles to outperform traditional machine learning models (e.g., XGBoost Chen & Guestrin (2016)) in downstream target data classification. Therefore, effective and generalizable context learning is necessary to achieve robust transfer learning across disparate domains. In computer vision, Seo et al. Seo et al. (2024) have proposed stochastic cross-attention (StochCA) learning using a pre-trained vision transformer (ViT). The StochCA method uses a pretrained ViT to obtain $key$ and $value$ representations for the target data. The scaled dot product between the $key$ representation from the pretrained ViT and the $query$ representation from the downstream ViT is used to compute the cross-attention between source and target. Although cross-attention works seamlessly for multi-source images with homogeneous feature representation, it is not trivial for a heterogeneous feature space across disparate tabular data sets. We propose learning a generalized context between domains using a $data-agnostic$ cross-attention mechanism. The proposed $data-agnostic$ cross-attention relies on pretrained attention weights rather than domain-specific $key$, $value$, and $query$ representations. The $data\text{-}agnostic$ cross-attention offers the generalized context required for cross-domain attention in tabular transfer learning (CATTLE). To our knowledge, this is the first paper to introduce generalized context learning for tabular data sets, especially for transfer learning across domains without shared features.

The key contributions of this work are as follows. First, this paper proposes one of the first transfer learning methods between tabular data sets of two distinct domains without requiring shared features. Second, a novel method for cross-attention is proposed using the projection weights of the transformer attention layer instead of using $key$, $value$, and $query$ feature representations. Third, cross-attention at the transformer weight level, rather than the feature level, yields generalized context suitable for $data\text{-}agnostic$ cross-domain transfer learning. Fourth, the proposed cross-attention by selective weight transfer in transformer layers yields a generalized context for the target data independent of the source-specific domain context and achieves state-of-the-art performance.

## 2 METHODOLOGY

### 2.1 PRELIMINARIES

A tabular data set $X \in \mathbb{R}^{N \times d}$ is structured in $N$ row samples and $d$ column features of varying types, where $d = a + b$ with $a$ categorical and $b$ numerical features. A transformer model treats each feature as a token and transforms the embedding ($E_i \in \mathbb{R}^{d_m}$) of the individual token $i$ into a context vector ($Z_i \in \mathbb{R}^m$), after learning between-feature attention as follows. The between-feature attention mechanism uses three trainable weights related to $query$, $key$, and $values$ ($W_q$, $W_k$, and $W_v$) of $\mathbb{R}^{m \times d_m}$ to project each token embedding ($E_i$) into its corresponding query ($q_i$), key ($k_i$), and value ($v_i$) vectors of $\mathbb{R}^{1 \times m}$, respectively, as shown in Equation 1.

$$q_i = E_i W_q, \quad k_i = E_i W_k, \quad v_i = E_i W_v \tag{1}$$

The attention score ($\omega_{ji}$) for the feature $j$ to the feature $i$ is obtained using the scaled dot product of the query vector of $j$ ($q_j$) with the key vector of $i$ ($k_i$), where $\omega_{ji} \neq \omega_{ij}$. Attention scores are normalized using the dimension of the key vector ($d_k$) and the softmax function to obtain the

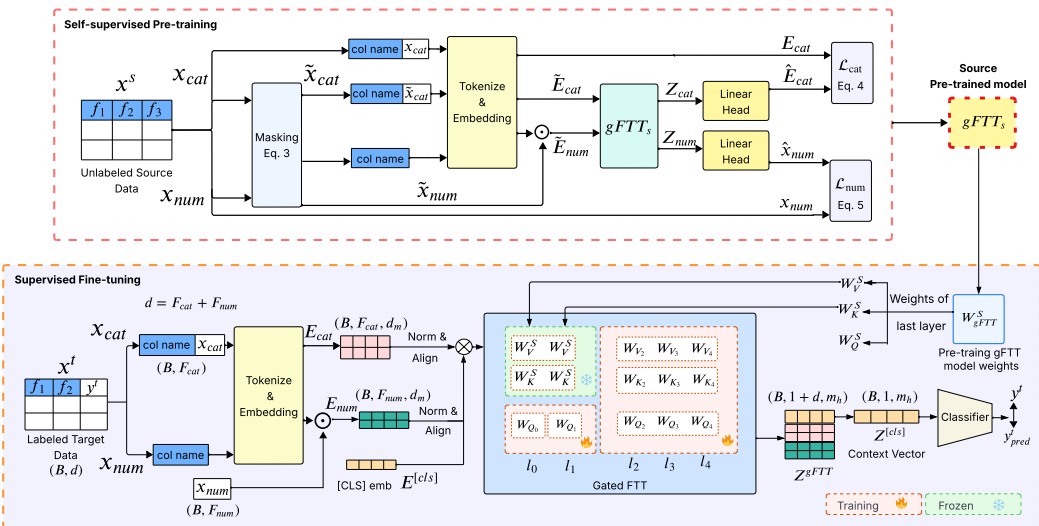

Figure 1: Overview of the CATTLE framework. Attention-related weights from a pretrained Gated Feature Tokenizer Transformer (gFTT) trained on source data are transferred to the downstream gFTT, which is then finetuned using target data. Emb. = Embedding, Cat. = Categorical, and Num. = Numerical features.

corresponding attention weight $\alpha_{ji}$ (Equation 2). The context vector ($Z_j$) of the query feature $j$ is the attention-weighted sum of the value vectors ($v_k$) corresponding to all features, as shown below.

$$\alpha_{ji} = softmax\left[\frac{(q_j^T k_i)}{\sqrt{d_k}}\right], Z_j = \sum_{k=1}^{d} \alpha_{jk} v_k \tag{2}$$

The context vector $Z_j$ encodes how feature $j$ focuses or attends to other features. Transformers with $h$ attention heads result in a context vector of $\mathbb{R}^{m_h}$, where $m_h = m \times h$.

## 2.2 PRETRAINING USING SOURCE DATA

Transfer learning involves pretraining on source data, followed by finetuning on target data to transfer knowledge. We obtain model pre-training separately by self-supervised and supervised learning for comparison. Self-supervised pre-training randomly masks a subset of data values and trains the model to reconstruct these values using the unmasked values. A binary mask $M \in \{0,1\}^{N \times d}$ identifies whether or not each data element $M_{ij}$ is to be masked. The masking decision is made independently for each feature value using a Bernoulli distribution parameter $P_{\text{mask}}$. Each mask entry $M_{ij}$ is set to 1 (masked) with probability $P_{\text{mask}}$, and 0 (unmasked) with probability $1 - P_{\text{mask}}$. The masking probability is set to $P_{\text{mask}} = 0.35$, which has been empirically shown to offer a good trade-off between reconstruction difficulty and the availability of contextual information in Ye et al. (2024). The value of a masked entry ($M_{ij} = 1$) is replaced with a [MASK] token, otherwise, the value is preserved, as shown in Equation 3.

$$\tilde{x} = x \odot (1 - M) + \text{[MASK]} \odot M \tag{3}$$

The operator $\odot$ denotes element-wise multiplication. The formulation in Equation 3 retains the original feature value when $M_{ij} = 0$, and substitutes it with the masking token [MASK]when $M_{ij} = 1$. The result is a partially observed input that preserves structure while forcing the model to reconstruct masked entries using the available context. To prevent complete information loss, we ensure that each sample has at least one unmasked feature. When all features in a row are masked, we explicitly unmask a fixed column in that row. The reconstruction of masked entries during pretraining is achieved differently for numerical and categorical features, as shown in Figure 1. The categorical feature name ("vehicle") and the value ("sedan") are tokenized to obtain the corresponding embeddings from a pretrained BERT model, which are then concatenated as a categorical feature embedding ($E^{cat}$). In contrast, the embedding of a numerical feature name is multiplied by the numerical value to obtain the embedding of the numerical feature ($E^{num}$). The embeddings for

the numerical ($E_{num}$) and categorical ($E_{cat}$) features are concatenated to represent individual samples for pre-training the Gated Feature-Tokenized Transformer (gFTT) Wang & Sun (2022). The reconstruction loss for categorical features is obtained using the cosine similarity between the reconstructed embedding vector ($\hat{E}_{ij}^{cat}$) and the input embedding $E_{ij}^{cat}$.

$$\mathcal{L}_{\text{cat}} = \sum_{i=1}^{n} \sum_{j=1}^{a} m_{ij} \cdot \left( 1 - \text{sim}(\hat{E}_{ij}^{cat}, E_{ij}^{cat}) \right) \tag{4}$$

Here, $m_{ij}$ represents an element of the binary mask matrix $M$, which is set to 1 to denote a masked value to be constructed, otherwise zero for unmasked values. The loss in Equation 4 maximizes the sum of similarity between the reconstructed and input embeddings of $a$ number of categorical features across $n$ samples. For numerical features, the model reconstructs the masked values ($\hat{x}_{ij}$) from the context vectors $Z$. The mean-squared reconstruction error is computed across the $n$ samples and $b$ numerical features as follows.

$$\mathcal{L}_{\text{num}} = \sum_{i=1}^{n} \sum_{j=1}^{b} m_{ij} \cdot \|x_{ij} - \hat{x}_{ij}\|^2 \tag{5}$$

The combined self-supervised feature reconstruction loss is obtained by averaging the number of features, as shown in Equation 6.

$$\mathcal{L}_{self} = \frac{1}{a}\mathcal{L}_{\text{cat}} + \frac{1}{b}\mathcal{L}_{\text{num}} \tag{6}$$

In supervised pretraining, the gFTT model is trained to predict the class labels of the source dataset using cross-entropy loss. We hypothesize that, for learning generalized context and obviating the need to annotate tabular data, self-supervised learning would be superior to supervised pretraining.

## 2.3 CROSS-ATTENTION FOR GENERALIZED CONTEXT

The $gFTT_s$ model trained by the source data set yields three transformer weights, $W_q^s$, $W_k^s$, and $W_v^s$, related to $query$, $key$, and $values$ representations, respectively. Here, $W_k^s$ and $W_v^s$ project the input to the $key\text{-}value$ (Equation 1) representation pair about the source data. However, the source data representation is assumed to be distributed across transformer layers, with the highest layer accumulating the most comprehensive knowledge. The layer with the most comprehensive knowledge is further assumed to have the most generalized context. Therefore, $W_k^s$ and $W_v^s$ of the highest layer of the pre-trained $gFTT_s$ are used as a foundation at the lowest layers of a new $gFTT_t$ (Figure 1). The new $gFTT_t$ is then trained using target data on the foundation of generalized context transferred from the pretrained model. The cross-attention in generalized context is achieved using the pretrained weights ($W_k^s$ and $W_v^s$) and projection weight ($W_q^t$) pertaining to the $query$ representation of target data. While $key$, $value$, and $query$ representations are specific to data domains, corresponding projection weights provide generalized context for *data-agnostic* cross-attention.

The downstream $gFTT_t$ model keeps the pretrained weights ($W_k^s$ and $W_v^s$) frozen to retain the generalized context at the foundation while updating all other weights in upper layers, propagating to the final inference for the target data set. It is worth noting that the hierarchical representation of images distributed across layers of convolutional neural networks is well known. However, a similar distribution of tabular data has not been well investigated for transformer layers. Therefore, we empirically investigate the distribution of knowledge across transformer layers for tabular data in this paper. Accordingly, the proposed *data agonistic* cross-attention is achieved by setting the general context foundation at the lowest two layers of $gFTT_t$ by the frozen weight pairs ($W_k^s$ and $W_v^s$) and simultaneously updating the $query$ weight $W_q^t$ using target data. Subsequently, the scaled dot product (Equation 2) between the $query$ and the $key$ representations of the target data, where $key$ is projected by the pre-trained $W_k^s$, yields cross-domain attention for transfer learning. The proposed method is presented in Algorithm 1.

# 3 EXPERIMENTS

## 3.1 TABULAR DATA IN CROSS-DOMAIN PAIRING

We use 14 tabular data sets from various application domains, such as health, finance, manufacturing, software testing, and industrial design, from the OpenML Vanschoren et al. (2013) repository. The

---

**Algorithm 1 Cross-ATtention Transfer LEarning (CATTLE)**

---

**Input:** Source data set $(X_s, y_s)$, Target data set $(X_t, y_t)$
**Model:** $gFTT_s$ **(source)**, $gFTT_t$ **(target)**
**Output:** Cross-attention $gFTT_{\text{CA}}$ for transfer learning
$gFTT_s$ **Pre-training: Source Data Set**
**for** epoch $= 1$ to $n\_epoch$ **do**
   **if** Pre-training $==$ Self-Supervised **then**
      $gFTT_s \leftarrow gFTT_s(X_s, \tilde{X}_s)$, Using Eq. 6
   **else**
      $gFTT_s \leftarrow gFTT_s(X_s, y_s)$
   **end if**
**end for**
$gFTT_s$ with $L$ attention layers $\{l_1, l_2, ..., L\}$
$\{W^{l_1}, ..., W^L\} \leftarrow gFTT_s$, Attention weights
$\{W_q^L, W_k^L, W_v^L\} \leftarrow W^L$, Last layer weights
$gFTT_t$ **Transfer Learning: Target Data Set**
Weight Replacement for Cross-Attention:
   $gFTT_t [W_k, W_v]^{\{\ell_0, \ell_1\}} \leftarrow [W_k^L, W_v^L]$
**for** epoch $= 1$ to $n\_epoch$ **do**
   $gFTT_{\text{CA}} \leftarrow gFTT_t(X_t, y_t)$ with $\{W_k, W_v\}_{Frozen}^{\{\ell_0, \ell_1\}}$
**end for**

---

data sets have varying mixes of numerical and categorical features, sample sizes ranging from 540 to 70000, and feature dimensions ranging from 8 to 76. A summary of the tabular data sets is presented in Table 2 in Appendix A.1. We select nine source and five target data sets to form ten source-target pairs for transfer learning. Unlike existing work on tabular transfer learning Ye et al. (2024); Zhu et al. (2023); Wang & Sun (2022), each pair of data sets is selected under a strict condition of disjoint feature space, ensuring that there are no shared features while differing in size and semantics.

## 3.2 BASELINE METHODS

Baseline methods are selected from two groups: 1) direct classification of the target data sets without transfer learning and 2) state-of-the-art transfer learning methods proposed for tabular data. The first group without transfer learning includes XGBoost and Logistic Regression (LR) as representative machine learning models, ResNet and MLP as standard DL methods, TabNet as a transformer with attention-based feature selection during training Arik & Pfister (2021), and FT-Transformer (FTT) using self-attention to learn tabular data representations Gorishniy et al. (2021). The state-of-the-art transfer learning methods include TransTab Wang & Sun (2022), XTab Zhu et al. (2023), and CM2 Ye et al. (2024). TransTab performs transfer learning between the source and target data sets with overlapping features by aligning shared embeddings Wang & Sun (2022). XTab utilizes multi-table pretraining to enhance generalization across diverse tabular data sets with minimal feature overlap Zhu et al. (2023). CM2 learns generalized feature representations by aligning statistical distributions across diverse and heterogeneous tables Ye et al. (2024).

## 3.3 MODEL IMPLEMENTATION AND EVALUATION

Transfer learning follows a two-step process: pre-training and fine-tuning. For CM2 and XTab, we use their pre-trained models trained on a large collection of tabular data sets. When pretrained models are unavailable, we use our source data sets for pre-training. In contrast, traditional machine learning and deep learning baselines are trained and evaluated using target data sets without transfer learning. The source and target data sets are randomly split into 70:10:20 for training, validation, and testing, respectively. Data with 70:10:20 splits are randomly sampled ten times using ten random seeds to facilitate statistical comparison of model performance. All models are validated using Optuna Akiba et al. (2019). In each of the 100 Optuna trials, a set of hyperparameter values is randomly sampled and evaluated on the validation set. The best model following the validation is used to report the target classification performance using the test data fold. Our proposed transfer learning approach uses a gated feature tokenized transformer (gFTT) with five layers, where each layer consists of

Table 1: Average area under the ROC curve (AUROC) and corresponding performance rank.

| | Logistic Regression | XGBoost | MLP | ResNet | FT-Transformer | TabNet | XTab | CM2 | TransTab | | CATTLE Supervised | CATTLE Self-Supervised |
|---|---|---|---|---|---|---|---|---|---|---|---|---|
| Target | | | | | | | Source | Source | Source | | | |
| DB | 0.822 (0.04) | 0.810 (0.04) | 0.798 (0.03) | 0.818 (0.03) | 0.788 (0.04) | 0.643 (0.03) | 0.834 (0.02) | 0.803 (0.03) | CD 0.799 (0.04) CG 0.818 (0.04) | | 0.803 (0.04) 0.808 (0.04) | 0.807 (0.04) 0.819 (0.04) |
| VH | 0.935 (0.01) | 0.925 (0.01) | 0.917 (0.01) | 0.855 (0.01) | 0.914 (0.01) | 0.794 (0.10) | 0.935 (0.01) | 0.893 (0.01) | MF 0.928 (0.01) DG 0.935 (0.01) | | 0.940 (0.01) 0.932 (0.01) | 0.933 (0.01) 0.942 (0.01) |
| CM | 0.703 (0.02) | 0.727 (0.02) | 0.704 (0.02) | 0.695 (0.03) | 0.724 (0.03) | 0.681 (0.08) | 0.721 (0.02) | 0.729 (0.02) | CH 0.726 (0.01) SK 0.733 (0.02) | | 0.721 (0.03) 0.726 (0.03) | 0.729 (0.02) 0.738 (0.02) |
| PC1 | 0.826 (0.04) | 0.834 (0.04) | 0.825 (0.06) | 0.681 (0.04) | 0.810 (0.04) | 0.843 (0.08) | 0.817 (0.05) | 0.845 (0.10) | CE 0.825 (0.04) SP 0.812 (0.04) | | 0.842 (0.05) 0.845 (0.04) | 0.840 (0.08) 0.813 (0.05) |
| CB | 0.816 (0.03) | 0.847 (0.04) | 0.603 (0.08) | 0.594 (0.07) | 0.677 (0.01) | 0.687 (0.05) | 0.822 (0.03) | 0.824 (0.05) | SP 0.827 (0.03) SB 0.825 (0.04) | | 0.829 (0.04) 0.832 (0.04) | 0.862 (0.04) 0.850 (0.04) |
| Avg. Rank | 5.0 (2.9) | 4.7 (1.6) | 8.1 (1.4) | 9.1 (3.0) | 8.5 (1.8) | 8.7 (3.5) | 4.6 (2.9) | 4.7 (3.2) | 4.8 (2.4) | | 3.9 (2.2) | 2.9 (2.4) |
| Overall Rank | 7 | 4 | 8 | 11 | 9 | 10 | 3 | 5 | 6 | | 2 | 1 |

(XTab Source: 52 AutoML Data sets; CM2 Source: Open Tabs 2000 Data sets)

eight attention heads. The feedforward network has a hidden layer of size 2048, ReLU activation, and dropout. The gFTT model hyperparameters remain at the default setting during pre-training. For transfer learning, the gFTT trained on the target data is tuned using Optuna. Classification performance is reported using accuracy and the area under the receiver operating characteristic (AUROC) curve. A win matrix is presented after statistically comparing the AUROC scores of a pair of methods using the Wilcoxon signed rank test and a significance level of $\alpha < 0.05$.

# 4 RESULTS

All experiments are performed on Ubuntu 22.04, powered by an Intel(R) Xeon(R) W-2265 CPU (24 logical cores) running at 3.70GHz, 64GB of RAM, and a Quadro RTX A4000 GPU with 16GB of video memory. The performance of the proposed CATTLE method is compared with the baselines using 1) the average rank based on AUROC scores and 2) the Wilcoxon signed rank statistical tests presented in a Win matrix.

## 4.1 MODEL TRAINING AND SELECTION

The upstream gFTT model with the default parameter settings is pre-trained on the source data. The self-supervised pre-training is performed for 1000 epochs with a batch size of 128 and a learning rate of 1e-4. The supervised pre-training uses 150 epochs, a batch size of 128, a learning rate of 3e-4, and a weight decay of 1e-2. The loss curves for self-supervised and supervised pre-training are presented in Figure 3(a) and 3(b), respectively. In contrast, the downstream gFTT model is tuned using the hyperparameter space presented in Table 3. Each Optuna trial randomly samples a set of hyperparameter values and evaluates them on validation data for 150 epochs. The convergence of training and validation losses for the target data set is demonstrated in Figure 3, specifically for self-supervised pretraining (Figures 3(c)) and supervised pretraining (Figure 3(d)).

## 4.2 TRANSFER LEARNING PERFORMANCE

Tables 1 and Table 4 in Appendix A.4 present the source-target data set pairs, average AUROC, and ACC scores after testing the target data sets. Transfer learning performance is then ranked based on AUROC and ACC scores. Traditional machine and state-of-the-art DL methods for tabular data are not designed for transfer learning and, therefore, are trained and tested using target data. Traditional machine learning methods, particularly XGBoost and logistic regression, consistently demonstrate strong performance across most data sets. For example, the performance rank of (AUROC: 4.7 (1.6), ACC: 5.0 (2.9)) is better than the best rank (AUROC: 8.1 (1.4), ACC: 6.6 (3.8)) among the state-of-the-art DL methods, including ResNet, MLP, FT-Transformer Gorishniy et al. (2021) and TabNet Arik & Pfister (2021). This observation supports a similar finding that traditional ML methods remain competitive with state-of-the-art DL methods Grinsztajn et al. (2022); Rabbani et al. (2024b) for tabular data. However, recent transfer learning methods for tabular data (XTab Zhu et al. (2023), CM2 Ye et al. (2024), and TransTab Wang & Sun (2022)) consistently rank better than their DL counterparts without transfer learning based on AUROC scores. The TransTab method consistently ranks the best among transfer learning baselines for tabular data based on ACC scores. A comparison with traditional yet competitive ML methods shows that transfer learning baselines (XTab Zhu et al. (2023), CM2 Ye et al. (2024)) consistently rank better in AUROC scores. When using the ACC score, TransTab outperforms all traditional ML methods. The proposed CATTLE with the supervised pretraining outperforms all baselines, including traditional ML, baseline DL, and transfer

Figure 2: Win matrix based on Wilcoxon signed-rank statistical tests. The cell value X/Y presents X as the number of times the row method statistically outperforms the column method. Y is the number of source–target data set pairs with statistically significant differences between a pair of methods.

learning methods proposed for tabular data. However, CATTLE with supervised pretraining (4.6 (2.7)) performs on par with TransTab (4.6 (3.0)) in terms of ACC scores. In general, CATTLE with self-supervised pretraining achieves the best performance rank (AUROC (2.9(2.4)) and ACC (2.4(1.5)), outperforming the second-best method, CATTLE with supervised pretraining, based on AUROC and ACC scores.

### 4.3 WIN MATRIX WITH STATISTICAL TESTS

The win matrix in Figure 2 shows the statistical comparison between two methods in a pair. The method with the statistically higher AUROC score wins in the pair. Each cell in the win matrix shows the number of source-target data sets in which one method wins over the other. The win matrix shows that CATTLE with self-supervised pre-training wins against all baseline methods on data sets that achieve statistical significance. For example, CATTLE with self-supervised pretraining and XGBoost produces statistically different AUROCs on four data sets, with CATTLE winning three out of four times. In contrast, CATTLE with supervised pretraining does not show statistically significant differences from TransTab (0/0). However, CATTLE with supervised pretraining wins all the significant cases against MLP (2/2), TabNet (7/7), and CM2 (2/2). XTab outperforms CATTLE supervised in a source-target data set pair (1/1).

### 4.4 ABLATION STUDY

Several factors can impact CATTLE performance. First, the distribution of knowledge across attention layers in gFTT is unknown for tabular data. Therefore, the selection of pretrained attention layers may impact downstream classification performance. Table 5 compares the transfer of different attention layers from the pretrained $gFTT_s$ to the downstream $gFTT_s$. The results suggest that the transfer of the weights of the top layer (gFTT$_s(W^L)$) to the two lowest layers of the downstream $gFTT_t$ (gFTT$_s(W^L) \rightarrow$gFTT$_t(W^{0,1})$) is the best choice for the proposed transfer learning. Second, the domain-specific context of the source data can affect the downstream classification of the target data set. Figure 4 shows that the difference between the maximum and minimum AUROC scores due to the varying source data sets is 0.018, suggesting a negligible impact of the context specific to the source data. The improved performance of CATTLE, despite the insignificant effect of domain-specific context, indicates the contribution of generalized context learned via cross-attention.

## 5 DISCUSSION

This article presents one of the first transfer learning methods for tabular data sets without shared features. Our research findings can be summarized as follows. First, cross-attention via transformer weights rather than $key$, $value$, and $query$ representations yields state-of-the-art performance in cross-domain transfer learning. Second, cross-attention at the transformer weight level yields generalized context independent of the upstream source context and representation. Third, a single data source can provide a *data-agnostic* generalized context for cross-domain learning without requiring large volumes of disjoint data sets from disparate domains. Fourth, self-supervised pretraining using unlabeled source data is better than its supervised counterpart for *data-agnostic* transfer learning. The key findings of this paper require further elucidation in the context of existing work.

The choice of a single-source data set over large volumes of multi-source data may contradict the general practice of pretraining models for transfer learning. Large language and vision models are developed using data from various application domains Radford et al. (2021). Thanks to shared image patterns and text semantics across domains, which facilitate seamless learning of general knowledge from large volumes of data Thwal et al. (2024). Similar shared features are not plausible between tabular data from medical records and banking transactions. Without proper inductive bias, training a model using disjoint data from disparate domains can compromise the integrity of knowledge due to domain conflicts. A similar perspective can explain why the pretrained CM2 model, despite being trained on 2000 tabular data sets, underperforms baselines using either a single (TransTab) or much fewer than 2000 data sets (CM2). Therefore, data-agnostic transfer learning, preferably using a generalized context similar to ours, is imperative and effective for tabular data.

The proposed *data-agnostic* learning minimizes cross-domain conflicts in two ways. First, the attention layer weights of the pretrained model are transferred to a new target model rather than fine-tuning the pretrained model downstream. Second, cross-domain attention is achieved using transformer weights instead of $key$ and $value$ representations. It can be argued that the core gFTT model for representation learning plays a major role in CATTLE performance. However, other baselines, including TransTab and CM2, use gFTT or a similar feature-tokenized transformer. Furthermore, selecting attention layers from pretrained gFTT models in our approach may not align with other transfer learning frameworks. In addition to the ablation study, we demonstrate the efficacy of five gFTT attention layers used in this study. Table 6 in Appendix A shows that the topmost layers of a pre-trained model are most effective for downstream classification performance. In line with this observation, the weights of the topmost pretrained attention layers are selected for downstream learning of target data in this paper. In the context of tabular data with limited samples, traditional machine learning is recommended, where transfer learning is not trivial due to the lack of effectively pretrained models. For example, the Cylinder Bands (CB) data set has the smallest sample size (540) and the highest number of features (39), with an even mix of numerical and categorical variables. Deep representation learning (ResNet, FT-Transformer) and even recent transfer learning methods (XTab, CM2, TransTab) fall short of the performance of the XGBoost classifier on the CB data set. However, CATTLE with self-supervised pre-training outperforms XGBoost, suggesting an effective solution to learning challenging tabular data with limited samples.

The proposed method is not without limitations, despite demonstrating state-of-the-art performance. The selection of attention layers for weight transfer is based on prior knowledge about knowledge distribution in deep layers, which would require a theoretical underpinning. More research is needed to optimize this framework by adopting a more systematic approach to selecting the attention layers. It is challenging to clearly explain the effectiveness of *data-agnostic* transfer learning without understanding the distribution of knowledge in multi-head attention layers.

## 6 CONCLUSIONS

This article presents a novel cross-domain attention method for the transfer learning of heterogeneous tabular data. Attention at the transformer weight level, rather than in $key$ and $value$ representations, yields *data-agnostic* generalized context necessary for cross-domain transfer learning. A single source data set can learn a generalized context for a downstream task in a different domain. The proposed generalized context in transfer learning outperforms state-of-the-art classification and transfer learning methods for tabular data, and even models pretrained on many data sets.

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

# A    APPENDIX

## A.1    SUMMARY OF SOURCE AND TARGET DATA SETS

Table 2: Summary of source and target data sets for transfer learning.

| Type | OpenML ID | Data set | Domain | Samples | Features | Numeric | Categorical | Classes |
|------|-----------|----------|--------|---------|----------|---------|-------------|---------|
| Source Data sets | 14 | mfeat-fourier (MF) | Shape Measurements | 2000 | 76 | 76 | 0 | 10 |
| | 31 | credit-g (CG) | Banking | 1000 | 20 | 7 | 13 | 2 |
| | 38 | sick (SK) | Thyroid Disease | 3772 | 29 | 7 | 22 | 2 |
| | 980 | optdigits (DG) | Optical Digits | 5620 | 64 | 64 | 0 | 2 |
| | 1504 | steel-plates-fault (SP) | Manufacturing | 1941 | 33 | 33 | 0 | 2 |
| | 40664 | car-evaluation (CE) | Car Pricing | 1728 | 21 | 0 | 21 | 4 |
| | 40701 | churn (CH) | Telecommunication | 5000 | 20 | 16 | 4 | 2 |
| | 45547 | cardiovascular-disease (CD) | Heart Disease | 70000 | 11 | 5 | 6 | 2 |
| | 45562 | seismic-bumps (SB) | Hazard Monitoring | 2584 | 18 | 14 | 4 | 2 |
| Target Data sets | 23 | cmc (CM) | Demographics | 1473 | 9 | 2 | 7 | 3 |
| | 37 | diabetes (DB) | Metabolic Disease | 768 | 8 | 8 | 0 | 2 |
| | 54 | vehicle (VH) | Automotive | 846 | 18 | 18 | 0 | 4 |
| | 1068 | pc1 (PC1) | Software Testing | 1109 | 21 | 21 | 0 | 2 |
| | 6332 | cylinder-bands (CB) | Industrial Design | 540 | 39 | 18 | 21 | 2 |

## A.2 OPTUNA CONFIGURATIONS

Table 3: Hyperparameter search space for all baselines and proposed methods.

| Parameter | Distribution |
|---|---|
| *XGBoost Chen & Guestrin (2016)* | |
| max_depth | UniformInt[1, 10] |
| learning_rate | LogUniform[$\exp(-7)$, 1] |
| n_estimators | UniformInt[100, 4000] |
| subsample | UniformFloat[0.2, 1.0] |
| colsample_bytree | UniformFloat[0.2, 1.0] |
| min_child_weight | LogUniform[$\exp(-16)$, $\exp(5)$] |
| gamma | LogUniform[$\exp(-16)$, $\exp(2)$] |
| reg_alpha | LogUniform[$\exp(-16)$, $\exp(2)$] |
| reg_lambda | LogUniform[$\exp(-16)$, $\exp(2)$] |
| *Logistic Regression Kleinbaum et al. (2002)* | |
| C | LogUniform[$1e^{-4}$, $1e^{2}$] |
| penalty | Categorical{"l1", "l2"} |
| max_iter | UniformInt[100, 1000] |
| *MLP, ResNet, FT-Transformer Gorishniy et al. (2021)* | |
| learning_rate | LogUniform[$1e^{-5}$, $3e^{-4}$] |
| batch_size | Int[32, 128] (step=32) |
| weight_decay | LogUniform[$1e^{-6}$, $1e^{-2}$] |
| hidden_dropout_prob | Categorical{0.0, 0.1, 0.2, 0.3, 0.4} |
| *TabNet Arik & Pfister (2021)* | |
| mask_type | Categorical{"entmax", "sparsemax"} |
| learning_rate | LogUniform[$1e^{-3}$, $1e^{-1}$] |
| cat_emb_dim | Int[8, 32] (step=8) |
| gamma | Uniform[1.0, 3.0] |
| batch_size | Categorical{64, 128, 256} |
| *XTab Zhu et al. (2023), TransTab Wang & Sun (2022), CM2 Ye et al. (2024)* | |
| learning_rate | LogUniform[$1e^{-5}$, $3e^{-4}$] |
| batch_size | Int[32, 128] (step=32) |
| weight_decay | LogUniform[$1e^{-6}$, $1e^{-2}$] |
| hidden_dropout_prob | Categorical{0.0, 0.1, 0.2, 0.3, 0.4} |
| warmup_ratio | Categorical{0.01, 0.05, 0.1} |
| *Proposed gFTT* | |
| learning_rate | LogUniform[$1e^{-5}$, $3e^{-4}$] |
| batch_size | Int[32, 128] (step=32) |
| weight_decay | LogUniform[$1e^{-6}$, $1e^{-2}$] |
| hidden_dropout_prob | Categorical{0.0, 0.1, 0.2, 0.3, 0.4} |
| warmup_ratio | Categorical{0.01, 0.05, 0.1} |

## A.3 Model Training Loss Curves

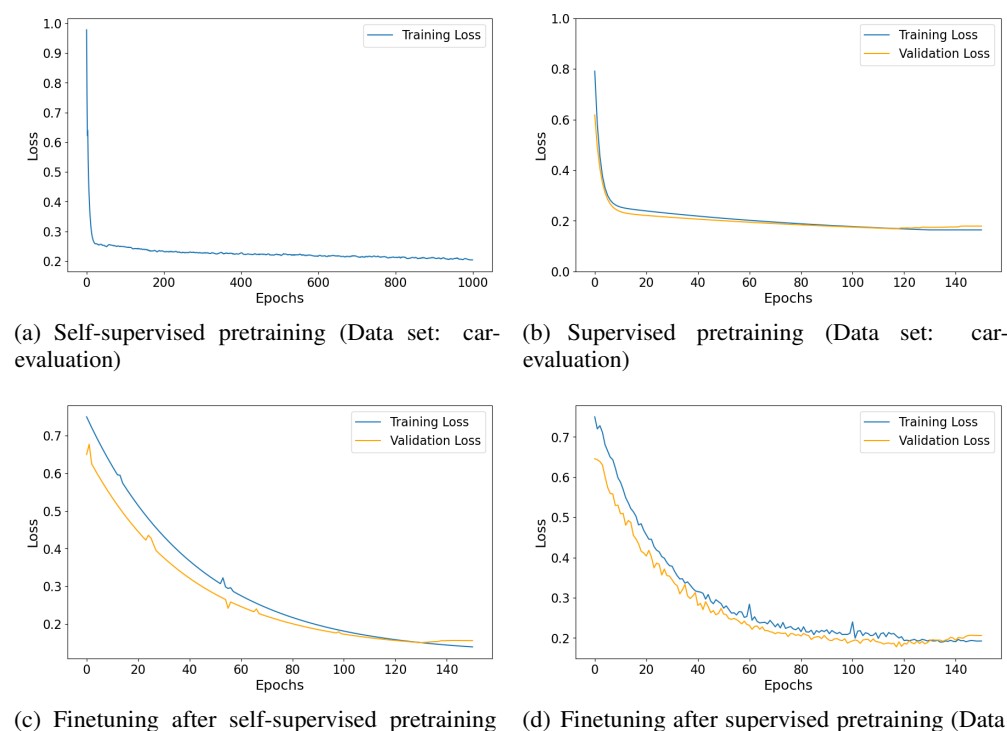

(a) Self-supervised pretraining (Data set: car-evaluation)

(b) Supervised pretraining (Data set: car-evaluation)

(c) Finetuning after self-supervised pretraining (Data set: pc1)

(d) Finetuning after supervised pretraining (Data set: pc1)

Figure 3: Transfer learning loss curves: (a) self-supervised pretraining, (b) supervised pretraining, (c) finetuning a self-supervised pretrained model using a target data set, and (d) finetuning a supervised pretrained model using a target data set.

## A.4 Additional Results

Table 4: Comparison of CATTLE with other baseline methods based on Accuracy (ACC).

| Target | Logistic Regression ACC | XGBoost ACC | MLP ACC | ResNet ACC | FT-Transformer ACC | TabNet ACC | XTab Source | XTab ACC | CM2 Source | CM2 ACC | TransTab Source | TransTab ACC | CATTLE Supervised ACC | CATTLE Self-Supervised ACC |
|---|---|---|---|---|---|---|---|---|---|---|---|---|---|---|
| DB | 0.747 (0.04) | 0.729 (0.04) | 0.753 (0.03) | 0.772 (0.03) | 0.754 (0.04) | 0.625 (0.05) | | 0.764 (0.02) | | 0.734 (0.02) | CD | 0.734 (0.04) | 0.735 (0.02) | 0.740 (0.04) |
| | | | | | | | | | | | CG | 0.735 (0.04) | 0.738 (0.05) | 0.746 (0.05) |
| VH | 0.736 (0.03) | 0.753 (0.03) | 0.754 (0.03) | 0.721 (0.04) | 0.767 (0.06) | 0.574 (0.01) | | 0.753 (0.02) | | 0.664 (0.02) | MF | 0.776 (0.03) | 0.774 (0.02) | 0.786 (0.02) |
| | | | | | | | 52 AutoML | | OpenTabs | | DG | 0.766 (0.03) | 0.758 (0.02) | 0.773 (0.02) |
| CM | 0.512 (0.04) | 0.551 (0.03) | 0.526 (0.02) | 0.536 (0.04) | 0.583 (0.06) | 0.766 (0.01) | Benchmark | 0.653 (0.03) | 2000 Data | 0.552 (0.03) | CH | 0.510 (0.02) | 0.534 (0.04) | 0.640 (0.03) |
| | | | | | | | Data sets | | sets | | SK | 0.558 (0.03) | 0.542 (0.04) | 0.647 (0.03) |
| PC1 | 0.829 (0.04) | 0.889 (0.01) | 0.822 (0.01) | 0.630 (0.01) | 0.710 (0.04) | 0.743 (0.04) | | 0.717 (0.05) | | 0.765 (0.10) | CE | 0.918 (0.04) | 0.931 (0.05) | 0.928 (0.01) |
| | | | | | | | | | | | SP | 0.925 (0.01) | 0.932 (0.06) | 0.929 (0.01) |
| CB | 0.746 (0.03) | 0.806 (0.04) | 0.673 (0.06) | 0.700 (0.05) | 0.557 (0.02) | 0.577 (0.04) | | 0.722 (0.03) | | 0.727 (0.05) | SP | 0.778 (0.04) | 0.758 (0.04) | 0.789 (0.04) |
| | | | | | | | | | | | SB | 0.764 (0.04) | 0.743 (0.07) | 0.778 (0.04) |
| Average Rank | 6.8 (2.5) | 5.0 (2.9) | 7.0 (2.2) | 8.1 (3.0) | 6.6 (3.8) | 7.9 (4.1) | | 5.6 (2.9) | | 7.3 (1.9) | | 4.6 (3.0) | 4.6 (2.7) | 2.4 (1.5) |
| Overall Rank | 7 | 4 | 8 | 11 | 6 | 10 | | 5 | | 9 | | 3 | 2 | 1 |

## A.5 ABLATION RESULTS

Table 5: Comparison of different combinations of pretrained gFTT layer weights in cross-attention transfer learning across source–target tasks.

| Source → Target | $\mathbf{gFTT}_s(W^L) \to \mathbf{gFTT}_t(W^0)$ | $\mathbf{gFTT}_s(W^L, W^{L-1}) \to \mathbf{gFTT}_t(W^{0,1})$ | $\mathbf{gFTT}_s(W^L) \to \mathbf{gFTT}_t(W^{0,1})$[Proposed] |
|---|---|---|---|
| credit-g → diabetes | 0.809 (0.03) | 0.803 (0.03) | **0.819 (0.04)** |
| optdigits → vehicle | 0.935 (0.02) | 0.917 (0.02) | **0.942 (0.01)** |
| sick → cmc | 0.746 (0.02) | 0.732 (0.02) | **0.738 (0.02)** |

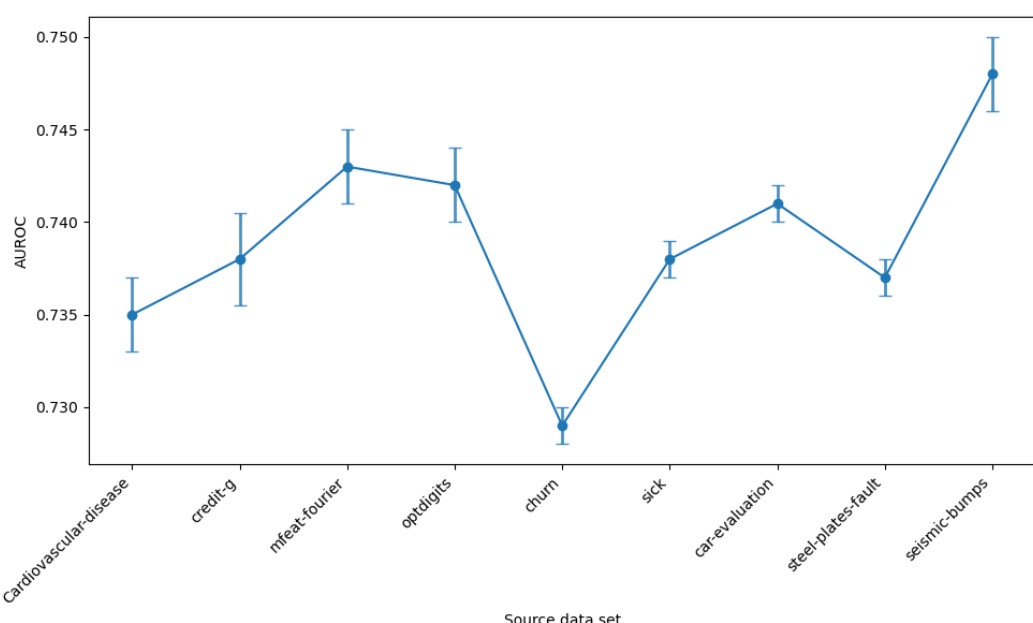

Figure 4: Effects of individual source data sets on the classification of the downstream CMC target data set. A source data set is used for self-supervised pretraining of the upstream model.

Table 6: Effects of the number of active attention layers of the pretrained gFTT model (supervised) on downstream target data classification (AUROC scores).

| Data set pairs | Top 4 Layers | Top 3 Layers | Top 2 Layers | Top 1 Layer |
|---|---|---|---|---|
| CD → DB | 0.818(0.051) | 0.814(0.055) | 0.810(0.060) | 0.821(0.057) |
| CE → PC1 | 0.787(0.071) | 0.810(0.051) | 0.833(0.047) | 0.831(0.035) |
| CH → CM | 0.735(0.046) | 0.732(0.039) | 0.741(0.043) | 0.743(0.039) |
| CG → DB | 0.803(0.063) | 0.808(0.059) | 0.806(0.063) | 0.804(0.061) |
| MF → VH | 0.932(0.014) | 0.938(0.014) | 0.929(0.015) | 0.927(0.012) |
| DG → VH | 0.928(0.010) | 0.929(0.010) | 0.926(0.010) | 0.923(0.012) |
| SK → CM | 0.753(0.044) | 0.755(0.044) | 0.750(0.037) | 0.745(0.047) |
| SP → CB | 0.842(0.042) | 0.846(0.029) | 0.841(0.040) | 0.838(0.032) |
| SP → PC1 | 0.814(0.041) | 0.805(0.044) | 0.820(0.044) | 0.846(0.032) |

## A.6 COMPUTATIONAL TIME

Table 7: Comparison of runtime and classification rank for transfer and non-transfer learning methods. For non-transfer methods, only training and inference times on the target data set are reported. For transfer learning methods, 'X' indicates usage of publicly released pretrained models for finetuning.

| Type | Method | Pretraining Runtime | Finetuning Runtime | Training Runtime | Inference Runtime | Total Runtime | Classification Rank (Std) |
|------|--------|--------------------|--------------------|------------------|-------------------|---------------|---------------------------|
| Non-transfer Learning | Logistic Regression | - | - | 7 | 1 | 8 | 5.0 (2.9) |
| | XGBoost | - | - | 8 | 1 | 9 | 4.7 (1.6) |
| | MLP | - | - | 21 | 1 | 22 | 8.1 (1.4) |
| | ResNet | - | - | 23 | 1 | 24 | 9.1 (3.0) |
| | FT-Transformer | - | - | 31 | 1 | 34 | 8.5 (1.8) |
| | TabNet | - | - | 12 | 1 | 13 | 8.7 (3.5) |
| Transfer Learning | XTab | X | 34 | - | 1 | 35 | 4.6 (2.9) |
| | CM2 | X | 42 | - | 1 | 43 | 4.7 (3.2) |
| | TransTab | 4608 | 44 | - | 1 | 4652 | 4.8 (2.4) |
| | CATTLE supervised | 5682 | 64 | - | 1 | 5746 | 3.9 (2.2) |
| | CATTLE self-supervised | 7702 | 87 | - | 1 | 7789 | 2.9 (2.4) |

