# OpenReview forum: "Cross-domain Attention for Transfer Learning between Tabular Data without Shared Features"
_ICLR.cc/2026/Conference — Submitted to ICLR 2026_

### Official Review · Reviewer_vqS8 · 2025-10-22

**Soundness:** 2
**Presentation:** 2
**Contribution:** 1
**Rating:** 2
**Confidence:** 5

**Summary:**

This paper proposes CATTLE, a data-agnostic Cross-domain Attention Transfer Learning framework for tabular data. Specifically, CATTLE reuses the attention-related weights from a pretrained Gated Feature Tokenizer Transformer trained on source data, for fine-tuning the target data. The authors evaluated their method on 10 pairs of source-target datasets, arguing the state-of-the-art performance.

**Strengths:**

1. The paper is well written.

2. Motivation is great. Transfer learning for tabular data is non-trivial and challenging problem, where its potential is great for improving the prediction performance of tabular tasks.

**Weaknesses:**

1. How did the authors choose the source dataset and target dataset? Can the authors provide the criteria? Because, it might be seen as cherry picked results because the target data set has a very small number of test data, leading to high variance of the score.

2. In addition, how are the source and target datasets paired? Why only the two source data is used for one target data? Is it possible to use all the source data at once for the pretraining? In this sense, it is also possible to use 2000 datasets in OpenTabs too.

3. I think the method is not that new compared to XTab, CM2, or TransTab. Architecture modification is just a marginal contribution, and many previous works follow a pretraining-finetuning framework. Where to freeze (or not freeze) seems to be just a design choice.

4. Lack of baselines. The authors only use the simple tree-based or deep-learning models as a baseline. These days, more powerful models are proposed, like TabPFNv2, ReaMLP, and so on. The authors should provide a comparison result against them.

5. Missing citations. There are other transfer learning frameworks for tabular data, for example, P2T [1]. This work also proposed to use a source data that has no overlapping columns with the target dataset.

6. Lack of ablation study. The authors should provide the impact of number of source data, some design choices like freezing the attention weight.

[1] Nam et al., Tabular Transfer Learning via Prompting LLM, COLM 2024.

**Questions:**

See Weaknesses.

---

> ### Author Response · Authors · 2025-11-20
>
> ## Weaknesses:
> > How did the authors choose the source dataset and target dataset? Can the authors provide the criteria? Because, it might be seen as cherry picked results because the target data set has a very small number of test data, leading to high variance of the score.
>
> Thank you! We explain the data selection criteria in Lines 292-297. Unlike vision, language, and graph models, there are no standard benchmarking data sets for Tabular models.  We selected the source–target pairs according to the criteria that each pair must have completely disjoint columns and be from distinct application domains, and that the pairs span a wide range of row and column sizes.  This paper is the first of its kind to evaluate cross-domain transfer learning on disjoint data sets. The choice of a limited data set is to demonstrate that “data-agnostic” cross-attention learning can be achieved using a single source data set. The low variance in Figure 4 (Appendix) shows that it is not the semantics of the source data that are used for transfer learning, but the generalized context learned from the transformer weights (regardless of the source data set).
>
> > In addition, how are the source and target datasets paired? Why only the two source data is used for one target data? Is it possible to use all the source data at once for the pretraining? In this sense, it is also possible to use 2000 datasets in OpenTabs too
>
> We have revised the Discussion Section (from Line 450). Two source data sets allow us to compare the effect of different sources on a single target data set.  While trivial in vision and language models (data from different sources always have shared patterns and semantics), a model pretrained on health records tabular data (blood pressure, cholesterol) and banking transaction data (credit scores) does not offer a plausible or effective way to combine the knowledge of two disparate domains in a single pretrained model. This problem is unique to tabular data sets, where training with multiple disparate and disjoint source data sets, creates domain conflicts and suboptimal performance. This explains why the CM2 model trained on 2000 Opentabs datasets underperforms models trained on a single source dataset (TransTab, CATTLE).
>
> > I think the method is not that new compared to XTab, CM2, or TransTab. Architecture modification is just a marginal contribution, and many previous works follow a pretraining-finetuning framework. Where to freeze (or not freeze) seems to be just a design choice.
>
> Our contribution is in the algorithm (Page 6), not in the architecture. Similar to our approach, XTab and TransTab used the same gFTT encoder to extract tabular embedding, but their underlying algorithms are distinct. As listed in contributions (Lines 139-145), our method is novel compared to XTab, CM2, and TransTab, because -
> * We achieve cross-attention at the weight level, not at the feature representation level (Q, K, V) like others,
> * Our attention is learned from generalized context, not specific to a domain context, and
> * Our pretraining and finetuning framework are not performed on the same model, but engineered for two different models,
> * Most importantly, our method is custom designed for data set pairs without shared features, whereas all other methods selected tabular data set pairs with common columns.
>
> > Lack of baselines. The authors only use the simple tree-based or deep-learning models as a baseline. These days, more powerful models are proposed, like as TabPFNv2 and ReaMLP. The authors should provide a comparison result against them.
>
> Unfortunately, we were not aware of these recent models, as our focus was on transfer learning methods under a unique data scenario rather than classifier benchmarking. We did not consider TabPFN due to the following statement in the paper, “TabPFN excels in handling small- to medium-sized datasets with up to 10,000 samples. For larger datasets, approaches such as CatBoost, XGB are likely to outperform TabPFN.", which suggests that XGB baselines are still competitive for tabular data sets. Our quick test results did not reveal superior performance compared to RealMLP.
>
> > Missing citations. There are other transfer learning frameworks for tabular data, for example, P2T [1]. This work also proposed to use a source data that has no overlapping columns with the target dataset.
>
> Cited P2T in line 56. Notably, the LLM prompt of P2T includes target data columns plus source columns correlated with target labels. Therefore, P2T source-target alignment is established using some common correlated columns (features), unlike ours.

---

> > ### Author Response · Authors · 2025-11-20
> >
> > > Lack of ablation study. The authors should provide the impact of number of source data, some design choices like freezing the attention weight.
> >
> > New Table 6 in the Appendix shows the impact of freezing the attention weights at different layers. The baseline model (XTab) is pretrained on 52, and CM2 is pre-trained on 2000 tabular datasets, whereas superior TransTab and CATTLE are trained on a single dataset. Despite pretraining on a large number of source data sets, XTab and CM2 model performances were not superior compared to that of single-data-set models.

---

### Official Review · Reviewer_99Sa · 2025-10-31

**Soundness:** 2
**Presentation:** 2
**Contribution:** 2
**Rating:** 2
**Confidence:** 3

**Summary:**

This paper proposes CATTLE(Cross-domain Attention Transfer Learning), a novel transfer learning framework for tabular data. Unlike image or text data, transfer learning for tabular data poses challenges due to the heterogeneity of features (type, structure, semantics) across different domains. While existing works often assume shared features between source and target datasets, this paper targets transfer learning between domains with entirely disjoint feature spaces. CATTLE selectively extracts only the Key (W_k) and Value (W_v) weights from the last attention layer of a source model (gFTT_s). These extracted weights are then injected into the first two layers of a new target model (gFTT_t) and frozen, while the target model only learns the Query (W_q) weights. The paper describes this mechanism as data-agnostic cross-attention, transferring knowledge at the weight level rather than the data representation level. Experiments on 10 disjoint dataset pairs (based on OpenML) show that CATTLE achieves strong performance rankings based on AUROC and ACC compared to existing SOTA models like XGBoost, TransTab, CM2, and XTab, with the self-supervised pretraining version performing better than the supervised one. The authors claim that CATTLE is effective even with a single source dataset, enabling transfer learning between tables without shared features and without requiring large-scale pretraining.

**Strengths:**

- This paper clearly defines and proposes a solution (CATTLE) for an important challenge in the tabular data domain. It addresses transfer learning between domains that have entirely different feature spaces and no shared features. While existing methods often assume some degree of feature overlap or focus on within-domain transfer, CATTLE overcomes these limitations and provides a more general approach. This improvement enhances the practical utility of tabular data, which is one of the most widely used data types across applications.

- The strategy of freezing the transferred Key (W_k) and Value (W_v) weights in the target model's initial layers while only training the Query (W_q) weights with target data is effective. This allows the model to retain the source domain's way of representing and summarizing information (Key/Value) while learning how to utilize and query this information specifically for the target domain's data (Query). This provides flexibility, preserving generalizable knowledge from the source while adapting to target domain characteristics.

- The finding that self-supervised pretraining using unlabeled source data achieves better performance than the supervised one is noteworthy. Acquiring large labeled datasets for tabular data is often challenging and costly, unlike image or text data. The effectiveness of CATTLE's self-supervised approach (masked feature reconstruction) suggests that model performance can be significantly improved using readily available large unlabeled datasets, making it a highly practical and powerful advantage for real-world applications

**Weaknesses:**

- The paper lacks enough explanations for why transferring attention weights (W_k, W_v) from the upper layers of the source model to the lower layers of the target model is the most effective approach. While the ablation study in Section 4.4 (Table 5) empirically supports this choice, the paper does not offer solid rationale for why this transfer is effective. There is insufficient analysis on what specific knowledge from the source domain is actually encoded in the transferred W_k, W_v weights and how they map to or interact with the features of the target domain.

- While CATTLE Self-Supervised achieves the top average rank in Table 2, the absolute performance differences compared to the second-best or other top-performing models are often very small. For instance, on the VH dataset, CATTLE Self-Supervised (0.942 AUROC) shows only a 0.007 improvement over TransTab (0.935) and XTab (0.935). Such minimal margins make it difficult to argue for a strong practical advantage derived from CATTLE's novel mechanism, in particular considering potential implementation cost of CATTLE. In addition, the comparisons are primarily made against models like TabNet and TransTab, which are not the most recent model, so it is unclear how CATTLE would perform relative to newer baselines.

- The paper claims in the Ablation Study (Section 4.4, Figure 4) that the choice of source dataset has a negligible impact, citing a AUROC difference of approximately 0.019 across different source datasets for the target cmc dataset. This interpretation appears contradictory to the significance attributed to smaller performance gains in Table 2, where a 0.007 AUROC difference on the VH dataset is implicitly used to argue for CATTLE's superiority over SOTA methods. This suggests an inconsistent standard for evaluating the importance of performance differences.

- The method’s generalizability has not been sufficiently validated. CATTLE is tested only on the gFTT architecture, and its performance on other tabular deep learning architectures such as TabNet is unknown. In addition, the paper does not quantify or analyze how different the source and target domains are. There is no examination of how CATTLE performs under various levels of domain shift, so it remains unclear whether the method is robust when the domains are highly dissimilar.

**Questions:**

- The CATTLE approach involves transferring and freezing specific attention weights (W_k, W_v) and training only part of the model, instead of fine-tuning the entire model. Theoretically, this could offer significant advantages in terms of computational cost and training speed during the fine-tuning phase compared to standard full-model fine-tuning approaches. It would be valuable to know whether this potential efficiency gain was quantitatively measured or compared in the experiments. For instance, are there comparative results on fine-tuning time or resource usage for CATTLE versus other baseline models on the same target dataset? If this potential efficiency gain is considered a key advantage of the methodology, what level of improvement could be expected?

- This study primarily validates CATTLE's performance using standard benchmark datasets like OpenML. While widely used, these datasets might differ significantly in scale and complexity from large-scale tabular data found in real-world web or industrial applications. It would be useful to clarify whether CATTLE's cross-attention mechanism and selective weight transfer are expected to demonstrate similar performance improvements, especially compared to SOTA models, on much larger and more complex tables. From a scalability perspective, potential limitations to CATTLE's performance or training stability as the dataset size increases could also be discussed, along with any plans for further experiments to validate its applicability on large-scale datasets.

- The paper claims to learn generalized representations even across entirely different domains without shared features, presenting AUROC and ACC results as evidence. However, these metrics primarily measure the final model's predictive performance on the target task and have limitations in directly showing how well the learned intermediate representations themselves have generalized or captured domain-invariant properties. It would be important to clarify whether AUROC/ACC alone are considered sufficient to conclude that CATTLE successfully learns generalized representations, and what the justification is for this interpretation. Alternatively, further analyses such as linear probing, domain discrimination tests, or representation visualization might be necessary or planned to more directly assess the quality and generalizability of the learned representations.

---

> ### Author Response · Authors · 2025-11-20
>
> ## Weakness
> > The paper lacks enough explanations for why transferring attention weights ($W_k$, $W_v$) from the upper layers of the source model to the lower layers of the target model is the most effective approach. While the ablation study in Section 4.4 (Table 5) empirically supports this choice, the paper does not offer solid rationale for why this transfer is effective. There is insufficient analysis on what specific knowledge from the source domain is actually encoded in the transferred $W_k$, $W_v$ weights and how they map to or interact with the features of the target domain.
>
> Thank you! We have rewritten Section 2.3 on page 5 to explain the choice. In conventional transfer learning, the upper layers of a pretrained model are usually kept active (selected) for downstream fine-tuning, while the lower layers are frozen. Our pretrained weights are selected accordingly. However, for disjoint data sets, the same pretrained model is not finetuned due to a lack of shared semantics; instead, a separate model must be chosen for the target data set. In that new model, the upper layer must learn target data-specific semantics, whereas the lowest two layers provide foundational weights for generalized context. Hence, those lowest-layer weights in the downstream (generalized context from the pretrained model) must be kept frozen, as it is done in practice. Notably, transformer weights ($W_k$, $W_v$, and $W_q$) are interpreted differently from conventional neural network weights or convolutional image filters. Tabular data with mixed types does not present the hierarchical structure that images do, as evident in the new Table 6 in the Appendix. While vision and language attentions can be meaningfully visualized to understand which specific knowledge is transferred, similar approaches are not trivial for tabular data with mixed data types (numerical, categorical, ordinal).
>
> > While CATTLE Self-Supervised achieves the top average rank in Table 2, the absolute performance differences compared to the second-best or other top-performing models are often very small. For instance, on the VH dataset, CATTLE Self-Supervised (0.942 AUROC) shows only a 0.007 improvement over TransTab (0.935) and XTab (0.935). Such minimal margins make it difficult to argue for a strong practical advantage derived from CATTLE's novel mechanism, in particular considering potential implementation cost of CATTLE. In addition, the comparisons are primarily made against models like TabNet and TransTab, which are not the most recent model, so it is unclear how CATTLE would perform relative to newer baselines.
>
> Statistical comparisons of model performance are presented in Figure 2 (Win Matrix), which shows more general performance than on a single dataset.  It is important to note that, unlike vision and language models, there is no single model that performs best across all datasets or scenarios in tabular data research (Line 48), due to heterogeneity in data types, statistics, structures, and domain semantics. For example, TabPFN is an SOTA model that excels only on small to mid-sized tabular data. Therefore, the average performance rank is always reported on tabular leaderboards along with statistical tests (Figure 2, Win Matrix) to compare model performance.
>
> > The paper claims in the Ablation Study (Section 4.4, Figure 4) that the choice of source dataset has a negligible impact, citing a AUROC difference of approximately 0.019 across different source datasets for the target cmc dataset. This interpretation appears contradictory to the significance attributed to smaller performance gains in Table 2, where a 0.007 AUROC difference on the VH dataset is implicitly used to argue for CATTLE's superiority over SOTA methods. This suggests an inconsistent standard for evaluating the importance of performance differences.
>
> That negligible difference in AUROC will not contribute to statistical significance in our Win Matrix (Figure 2). In our comparison, we use the average performance rank across 10 data set pairs, even though CATTLE may not be significantly superior in all 10 pairs (no model is an absolute winner in tabular research – “no free lunch” theory in machine learning). For example, a tabular dataset that achieves 99% accuracy with simple logistic regression has a linear decision boundary. In this case, other deep learning and non-linear classifiers are likely to overfit and result in poor performance.

---

> > ### Author Response · Authors · 2025-11-20
> >
> > > The method’s generalizability has not been sufficiently validated. CATTLE is tested only on the gFTT architecture, and its performance on other tabular deep learning architectures such as TabNet is unknown. In addition, the paper does not quantify or analyze how different the source and target domains are. There is no examination of how CATTLE performs under various levels of domain shift, so it remains unclear whether the method is robust when the domains are highly dissimilar.
> >
> > gFTT is a unified tabular data encoder, which is also used in the most recent transfer learning baselines – Xtab and TransTab. For fair benchmarking of algorithms (main contribution – Page 6), the architecture should remain unchanged. SOTA models are expected to be built on SOTA encoders. TabNet ranks 10th in our results, and using it as a tabular data encoder would not provide a quality embedding necessary for an SOTA model. Our selected source-target domains (Table 1- Appendix) chosen are highly dissimilar, with no common columns – e.g.,  “banking” versus “metabolic disease” data sets. To our knowledge, no prior tabular data transfer learning papers have considered such dissimilar pairs of disjoint data sets.
> >
> > ## Questions
> > > The CATTLE approach involves transferring and freezing specific attention weights (W_k, W_v) and training only part of the model, instead of fine-tuning the entire model. Theoretically, this could offer significant advantages in terms of computational cost and training speed during the fine-tuning phase compared to standard full-model fine-tuning approaches. It would be valuable to know whether this potential efficiency gain was quantitatively measured or compared in the experiments. For instance, are there comparative results on fine-tuning time or resource usage for CATTLE versus other baseline models on the same target dataset? If this potential efficiency gain is considered a key advantage of the methodology, what level of improvement could be expected?
> >
> > Yes, we have shared the computational time in the Appendix (Table 7).
> >
> > > This study primarily validates CATTLE's performance using standard benchmark datasets like OpenML. While widely used, these datasets might differ significantly in scale and complexity from large-scale tabular data found in real-world web or industrial applications. It would be useful to clarify whether CATTLE's cross-attention mechanism and selective weight transfer are expected to demonstrate similar performance improvements, especially compared to SOTA models, on much larger and more complex tables. From a scalability perspective, potential limitations to CATTLE's performance or training stability as the dataset size increases could also be discussed, along with any plans for further experiments to validate its applicability on large-scale datasets.
> >
> > Our experiences with real-world web and industrial data are –
> > * Such data include a large volume of unstructured text, unlike structured tabular data with mixed types.
> > * SOTA deep learning models converge to similar performance scores on databases with millions of rows after finetuning. In such a case, the performance difference is evident only at the zero-shot level. However, finetuning accuracy is always better than zero-shot. Such a large sample size does not pose the problem of learning with limited samples, which is quite common in tabular data domains. Complex tabular data, such as that in relational databases, needs complex graph-based models. Furthermore, our data-agnostic approach does not rely on (large) data to learn generalized context, unlike standard vision and language models.

---

> > > ### Author Response · Authors · 2025-11-20
> > >
> > > > The paper claims to learn generalized representations even across entirely different domains without shared features, presenting AUROC and ACC results as evidence. However, these metrics primarily measure the final model's predictive performance on the target task and have limitations in directly showing how well the learned intermediate representations themselves have generalized or captured domain-invariant properties. It would be important to clarify whether AUROC/ACC alone are considered sufficient to conclude that CATTLE successfully learns generalized representations, and what the justification is for this interpretation. Alternatively, further analyses such as linear probing, domain discrimination tests, or representation visualization might be necessary or planned to more directly assess the quality and generalizability of the learned representations.
> > >
> > > We agree. Discussion Section – from Lines 443 has been updated with further explanation and justification. However, intermediate representations in images (pixels) and text (words) can be visualized to explain the learning mechanism, whereas this is not trivial with tabular data of mixed data types. Alternatively, we show in Figure 4 (Appendix) demonstrates that downstream AUROC/ACC scores are negligibly affected by the source, substantiating the claim that the performance gain is due to the generalized context (independent of the source), not the source semantics.

---

### Official Review · Reviewer_ttcS · 2025-11-01

**Soundness:** 2
**Presentation:** 2
**Contribution:** 1
**Rating:** 2
**Confidence:** 5

**Summary:**

This paper proposes Cross-domain Attention Transfer Learning (CATTLE), a model that allows transfer learning between tabular datasets without shared features. The work is based on (1) self-supervised pre-training from source data, and (2) fine-tuning on specific dataset of interest. CATTLE extracts attention-related weights from a pretrained Gated Feature Tokenizer Transformer (gFTT) trained on source data are transferred into a newly initialized gFTT, which is then finetuned using target data. The experiments show solid performances compared to several baselines.

**Strengths:**

In general, the paper is easy to follow. The biggest strength of CATTLE would be the learning of the relationship between tabular data sets from different domains without requiring a common feature space, which can be useful in various settings of transfer learning and domain adaptation.

**Weaknesses:**

-	The paper needs to enrich the works in tabular learning for pre-training and transfer, such as TabPFN or CARTE.
-	The choice of source-target combination requires more justification.
-	The choice of hyperparameters space is limited (for instance for XGB).
-	The baselines should include more recent tabular learning models of TabPFNv2, and RealMLP.
-	The experiment results is limited. If CATTLE can cope with non-shared features, than it means that it can also cope with cases where there are limited matching of the shared features. There could be more datasets that can be explored to truly see the value of CATTLE.
-	The writings of the paper can be improved.
-	Results on smaller train-size on the target dataset may show effectiveness of CATTLE.

**Questions:**

-	What are the computational times for running the experiments?
-	What is the reason behind selecting only one source for the experiments?
-	What is the reason behind the presented combination of source-target datasets for the experiments?
-	What is the reason behind selecting BERT to extract the column features?
-	Through the recent advances in tabular learning, it seems possible to use llms to encode rows in a table as a sentence and perform various forms of domain adaption or transfer learning. Have the authors considered these as baselines?

---

> ### Author Response · Authors · 2025-11-20
>
> ## Weakness:
> > The paper needs to enrich the works in tabular learning for pre-training and transfer, such as TabPFN or CARTE.
>
> Thank you ! Unfortunately, we were not aware of these recent models. We did not consider TabPFN due to the following statement in the paper, “TabPFN excels in handling small- to medium-sized datasets with up to 10,000 samples. For larger datasets, approaches such as CatBoost, XGB are likely to outperform TabPFN.", which suggests that XGB baselines are still competitive for tabular data sets. CARTE is a graph-based model pretrained on large language data sets. Our models do not intend to leverage large language data that may have already been memorized (trained on) publicly available datasets. Given the timeframe, we were unfortunately unable to produce all the requested results.
>
> > The choice of source-target combination requires more justification.
>
> As mentioned in Lines (58-61, 109-110, 136–140), all transfer learning methods for tabular data assume data sets with shared or common features to facilitate alignment in semantics and representations. In reality, tabular datasets from disparate domains rarely share feature columns, which violates the assumption required by existing transfer learning approaches. Therefore, disjoint source and target data set pairs are chosen from different domains with no shared columns/features.
>
> > The choice of hyperparameters space is limited (for instance for XGB).
>
> We have expanded the hyperparameter space and updated the results in tables and figures.
>
> > The baselines should include more recent tabular learning models of TabPFNv2, and RealMLP.
>
> Unfortunately, these models were proposed recently because we focused on more recent transfer learning models (TransTab, Xtab, CM2). In our experiments, given this limited time, the proposed method did not significantly outperform RealMLP.
>
> > The experiment results is limited. If CATTLE can cope with non-shared features, than it means that it can also cope with cases where there are limited matching of the shared features. There could be more datasets that can be explored to truly see the value of CATTLE.
>
> Yes, when there are limited matching features, attention should be computed at the Key, Value, and Query levels to ensure proper semantic and feature alignment in transfer learning (similar to existing transfer learning methods). CATTLE is custom-built for disjoint tabular data sets only when domains are not comparable; it’s not a general or universal solution for all tabular data scenarios. Because learning disjoint tabular data without shared context has not been explored in the literature, we propose a solution for this particular data scenario.
>
> > The writings of the paper can be improved.
>
> As highlighted in multiple sections, we have rewritten and revised the papers to explain the methods and contributions better. The discussion section is entirely rewritten.
>
> > Results on smaller train-size on the target dataset may show effectiveness of CATTLE.
>
> Yes, as mentioned in Line 466, “For example, the Cylinder Bands (CB) data set has the smallest sample size (540) and the highest number of features (39), with an even mix of numerical and categorical variables. Deep representation learning (ResNet, FT-Transformer) and even recent transfer learning methods (XTab, CM2, TransTab) fall short of the performance of the XGBoost classifier on the CB data set. However, CATTLE with self-supervised pre-training outperforms XGBoost, suggesting an effective solution to learning challenging tabular data with limited samples.”

---

> > ### Author Response · Authors · 2025-11-20
> >
> > ## Questions
> >
> > > What are the computational times for running the experiments?
> >
> > We have included a new table with computational time in Appendix (Table 7).
> >
> > > What is the reason behind selecting only one source for the experiments?
> >
> > Lines 443-453 in the discussion have been updated. While trivial for images and text (where vision and language semantics are shared across domains), disjoint tabular data sets from disparate domains cannot effectively generalize a pretrained model. For example, A model trained on health records (blood pressure) will have its knowledge overridden my data related to banking transactions (credit scores). However, A generalized context should not be domain or data-dependent. Considering the two conditions above, we find that a single source data set is sufficient to learn the general context, e.g., how to project data using attention weights in a downstream learning task.
> >
> > > What is the reason behind the presented combination of source-target datasets for the experiments?
> >
> > We stated in Lines (58-61, 109-110, 136–140).  The reason is that source and target data sets must be disjoint, similar to many real-world scenarios, with no shared columns, and from different domains. Also, source datasets have more samples than the target datasets, as expected in conventional transfer learning.
> >
> > > What is the reason behind selecting BERT to extract the column features?
> >
> > BERT is a part of the unified tabular data encoder gFTT, a framework used in other baselines (TransTab, CM2) for extracting column embeddings. For fair benchmarking of algorithms, the encoder and model architectures are kept similar to the baselines. A superior encoder (e.g., Nomic) can be used to improve representation and boost performance, but this would mask the true contribution of the proposed algorithm.
> >
> > > Through the recent advances in tabular learning, it seems possible to use llms to encode rows in a table as a sentence and perform various forms of domain adaption or transfer learning. Have the authors considered these as baselines?
> >
> > LLMs are decoder models. Therefore, LLMs must be repurposed as encoder models. Also, LLMs expect contextual words in a sentence, and tabular-to-text transformation is a critical step that requires further work for effective domain adaptation and transfer learning. When tabular data are permutation invariant, sentence inputs to LLMs follow positional encoding. Also, LLM baselines are often trained on publicly available datasets we use for benchmarking and can be prone to data memorization.

---

### Official Review · Reviewer_GMLi · 2025-11-10

**Soundness:** 1
**Presentation:** 3
**Contribution:** 2
**Rating:** 2
**Confidence:** 4

**Summary:**

This paper proposes CATTLE (Cross-domain Attention Transfer Learning), a framework for transfer learning between tabular datasets with disjoint feature spaces. The method builds upon a modified FT-T architecture (Gated Feature Tokenizer Transformer, gFTT), and the key idea is to transfer and freeze the key/value projection matrices of the attention layers from a pretrained source model, while retraining the query projections on a new target dataset. The authors argue that this enables data-agnostic structural transfer between heterogeneous tables. Experiments on multiple OpenML datasets demonstrate improvements over classical baselines and prior tabular transfer methods such as TransTab, XTab, and CM2.

**Strengths:**

- The paper addresses a challenging and underexplored problem: transfer learning across heterogeneous tabular domains without shared features.

- The idea of transferring attention weights rather than feature-level representations is interesting.

**Weaknesses:**

- **Lack of theoretical foundation and over-reliance on empirical results.**
The proposed transfer mechanism is motivated almost entirely by empirical observation, without any theoretical or representational justification. Most architectural decisions—including which layers to transfer, which weights to freeze, and how to structure the transfer—are justified solely by ablation performance. Without an analysis of what these weights represent, it is unclear why they should generalize across feature spaces.

- **Unconvincing and counterintuitive “top-to-bottom” layer transfer.**
The practice of copying deep (semantic) layers from the source transformer into shallow (low-level) layers of the target transformer contradicts the conventional understanding of hierarchical representation learning. Deep layers typically capture domain-specific semantics, while shallow layers encode general patterns. Reversing this hierarchy lacks any conceptual rationale, raising doubts about the generality of the claimed “structural transfer.”

- **Contradiction between the use of column-level semantic embeddings and the absence of adaptive column matching.**
The model depends on explicit column embeddings and fixed feature order, which breaks permutation invariance and introduces implicit dependence on feature semantics. However, it simultaneously claims to be “data-agnostic” and performs no adaptive matching or semantic alignment between source and target columns. This creates an internal inconsistency: the method relies on column-level semantics to function, yet provides no mechanism to align or reconcile them across heterogeneous domains.

**Questions:**

- What is the conceptual basis for transferring deep source layers into shallow target layers?
How does this align with known representational hierarchies in transformers?

- Can the authors provide any evidence (e.g., attention visualization) showing that the transferred weights capture reusable relational patterns?

- Would the method’s behavior change if the target dataset’s column order were permuted?

- Is there any evidence that the transferred attention structure captures semantically meaningful relations rather than random initialization bias?

---

> ### Author Response · Authors · 2025-11-20
>
> ## Weakness:
>
> > Lack of theoretical foundation and over-reliance on empirical results. The proposed transfer mechanism is motivated almost entirely by empirical observation, without any theoretical or representational justification. Most architectural decisions—including which layers to transfer, which weights to freeze, and how to structure the transfer—are justified solely by ablation performance. Without an analysis of what these weights represent, it is unclear why they should generalize across feature spaces.
>
> Thank you. We have revised Section 2.3 on Page 5. The weights ($W_k$, $W_q$, $W_v$) represent the projection weights for the Key, Query, and Value representations (Eq. 1). The principles of computing attention provides our representational justification. While conventional feature-level (K, Q, V) representations assume shared semantics for attention learning, tabular data from disparate domains (e.g., medical versus banking) do not share a common feature space or context. Therefore, to establish a data-agnostic and generalized context, we use weights ($W_K^s$ and $W_V^s$) instead of their corresponding “Key-Value” representations of the source model.
>
> Intuitively, the key projection ($W_K^s$) is paired with the Query weights ($W_Q^t$) of the target model to establish cross-domain attention (Attention = Query*Key). In general, our cross-domain attention is computed at the attention-weight level rather than the feature-representation level to achieve the generalized context independent of source data representations. Similar intuitive freezing of “Key-Value” representation from the source model, we freeze the corresponding source pretrained weights ($W_K^s$, $W_V^s$) instead, which is not an arbitrary choice. Appendix (Table 6) shows that the downstream performance is largely insensitive to the selection of the layers because the weights from the top-most layer are sufficient for learning the generalized context, especially when we do not intend to align the semantics or representations of two disjoint tabular data sets.
>
> > Unconvincing and counterintuitive “top-to-bottom” layer transfer. The practice of copying deep (semantic) layers from the source transformer into shallow (low-level) layers of the target transformer contradicts the conventional understanding of hierarchical representation learning. Deep layers typically capture domain-specific semantics, while shallow layers encode general patterns. Reversing this hierarchy lacks any conceptual rationale, raising doubts about the generality of the claimed “structural transfer.”
>
> We have rewritten Section 2.3 on page 5 to explain the choice. In conventional transfer learning, the upper layers of a pretrained model are usually kept active (selected) for downstream fine-tuning, while the lower layers are frozen. Our pretrained weights are selected accordingly. However, for disjoint data sets, the same pretrained model is not finetuned due to a lack of shared semantics; instead, a separate model must be chosen for the target data set. In that new model, the upper layer must learn target data-specific semantics, whereas the lowest two layers provide foundational weights for generalized context. Hence, those lowest-layer weights in the downstream (generalized context from the pretrained model) must be kept frozen, as it is done in practice.
>
> The interpretation and hierarchy of traditional neural network weights, convolutional filters, and attention weights of the transformer layer (corresponding to key, value, and query) are distinct.  The conventional understanding of hierarchical representation learning applies to language (words) and vision (pixels) semantics, but it has not been intuitive and successful with tabular data. A model pretrained on medical records tabular data (e.g. cholesterol) is not expected to be finetuned (or share semantics) on banking transaction (credit scores) tabular data, which makes the foundation tabular data model so challenging. We added a new Table (Table 6 in the Appendix) to show that the pretrained attention layers do not exhibit a hierarchical distribution from shallow to deep layers, as commonly observed in vision and language models. The inductive bias for tabular data differs from that for images and text, and so does the hierarchy of data representation of tabular data.

---

> > ### Author Response · Authors · 2025-11-20
> >
> > > Contradiction between the use of column-level semantic embeddings and the absence of adaptive column matching. The model depends on explicit column embeddings and fixed feature order, which breaks permutation invariance and introduces implicit dependence on feature semantics.
> > However, it simultaneously claims to be “data-agnostic” and performs no adaptive matching or semantic alignment between source and target columns. This creates an internal inconsistency: the method relies on column-level semantics to function, yet provides no mechanism to align or reconcile them across heterogeneous domains.
> >
> > Our transformer model does not use any positional encoder or causal attention mask, which typically learn the feature order. The baseline models (TransTab, CM2) use the same unified tabular feature (column) encoder (gFTT), which has been tested for tabular data with permutation invariance.  Adaptive matching and semantic alignment are plausible when domains share semantics (common columns) that can be aligned adaptively at the feature-level. A model trained on natural images can be fine-tuned with medical images because images share general patterns across domains. In our context, banking data (credit scores) would not have a plausible context for medical records data (blood pressure) when computing attention via adaptive matching or semantic alignment. Our column (feature) embeddings are used for self-supervised training of the transformer weights, where weights are known to capture “a generalized projection mapping” independent of the data or domain context. Accordingly, the downstream attention is intuitively computed by pairing the “key” weight ($W_k$) of the source with the “query” weight ($W_q$) of the target, reconciling disparate domains at the weight level without explicitly aligning their feature representations (Key, Value, Query).
> >
> > ## Questions:
> >
> > > What is the conceptual basis for transferring deep source layers into shallow target layers? How does this align with known representational hierarchies in transformers?
> >
> > We have updated the entire discussion section (Lines 454-464) to elucidate this question. As detailed above, the deep layers of a pretrained model are usually finetuned, keeping the lowest layers frozen. Therefore, we selected the deep layers of the pretrained model for the downstream task on target data.
> >
> > Unlike images and text, tabular data sets do not reveal an explicit and explainable hierarchy of knowledge distributions across attention layers (Table 6 - Appendix). Particularly for tabular data, we demonstrate in Table 5 that the Key and Value projection weights (unlike neural network weights) of the deep source layer (the topmost layer) are sufficient to capture the general context at the attention-weight level. This is because we are not relying on the feature representation (semantics or alignment) in the downstream.
> >
> > > Can the authors provide any evidence (e.g., attention visualization) showing that the transferred weights capture reusable relational patterns?
> >
> > Thank you. Attention visualization is practical for explaining vision and language models because images and texts can be described (and recognized) by humans. For feature vectors with mixed data types (categorical, numerical) in tabular data, it can be counterintuitive to show how the blood pressure feature attends to the credit score features to yield a general context, especially when the cross-attention is not explicitly learned or aligned at the feature level.
> >
> > > Would the method’s behavior change if the target dataset’s column order were permuted?
> >
> > We independently tested the model's performance across varying column orders and did not observe any variation in predictive performance. There is no positional encoder and causal attention masks that are used to encode position or order of the features. We use a state-of-the-art unified tabular data encoder (a gated feature tokenized transformer) that has been tested for permutation invariance and has been adopted in similar research.
> >
> > > Is there any evidence that the transferred attention structure captures semantically meaningful relations rather than random initialization bias?
> >
> > Semantically meaningful relations can be visualized and explained in images (e.g., facial areas) or in text (verbs and nouns), but this is unfortunately not trivial for tabular data with mixed data types, especially when two disparate domains share no columns to establish meaningful alignment in semantics at the feature representation level.

---

### Meta-Review · Area_Chair_wycr · 2026-01-05

**Summary:**

Several reviewers express a lack of understanding of the motivations of the architectural choices, from a theoretical, or at least conceptual, standpoint. The empirical work is not very convincing, with limited and not up-to-date baselines (including some that are clearly relevant to the framing of transfer without feature correspondence, such as the CARTE line of work).

**Reviewer Concerns:**

The modifications made during the rebuttal to explain the architecture choices bid not quite clarify things to my eyes. The authors did not address the shortcoming of limited baselines in the rebuttal, in addition to jusitify not including TabPFN, they reference in the replies the limitations of the original TabPFN paper, published in 2022, limitations that have been addressed by the TabPFN2, and even more the TabPFN2.5 improvements.

**Reviewer Scores:**

I don't believe that the scores would have changed much, as the answers were not backed by much new evidence.

---

### Decision · Program_Chairs · 2026-01-26

Reject